# Modal-Enhanced Semantic Modeling for Fine-Grained 3D Human Motion Retrieval

## ABSTRACT

Text to Motion Retrieval (TMR) is an emerging task to retrieval relevant motion sequence with the nature language description. The existing dominant approach is to learn a joint embedding space to measure global-level similarities. However, simple global embeddings are insufficient to represent complicated motion and textual details, such as the movement of specific body parts and the coordination among these body parts. In addition, most of the motion variations occur subtly and locally, resulting in semantic vagueness among these motions, which further presents considerable challenges in precisely aligning motion sequences with texts. To address these challenges, we propose a novel Modal-Enhanced Semantic Modeling (MESM) method, focusing on fine-grained alignment through enhanced modal semantics. Specifically, we develop a prompt-enhanced textual module (PTM) to generate detailed descriptions of specific body part movements, which comprehensively captures the fine-grained textual semantics for precise matching. We employ a skeleton-enhanced motion module (SMM) to effectively enhance the model's capability to represent intricate motions. This module leverages a graph convolutional network to meticulously model the intricate spatial dependencies among relevant body parts. To improve the sensitivity to the subtle motions, we further propose a text-driven semantics interaction module (TSIM). The TSIM first assigns motion features into a set of aggregated descriptors, then employs the cross-attention to aggregate discriminative motion embeddings guided by text, enabling precise semantic alignment between subtle motions and corresponding texts. Extensive experiments conducted on two widely used benchmark datasets, HumanML3D and KIT-ML, demonstrate the effectiveness of our proposed method. Our approach outperforms existing state-of-the-art retrieval methods, achieving significant Rsum improvements of 24.28% on HumanML3D and 25.80% on KIT-ML.

## CCS CONCEPTS

• **Information systems** → **Retrieval models and ranking**; **Multimedia and multimodal retrieval**; **Novelty in information retrieval**.

## KEYWORDS

3D Human Motion, Text-to-Motion Retrieval, Semantic Modeling, Semantics Alignment

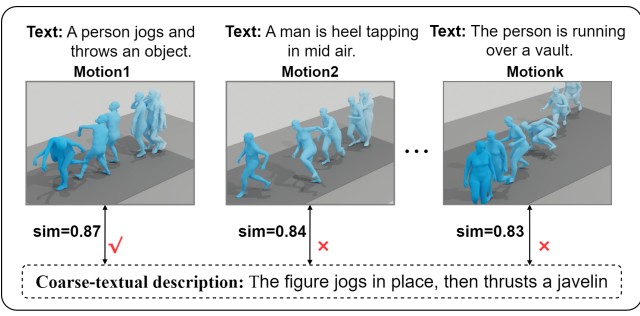

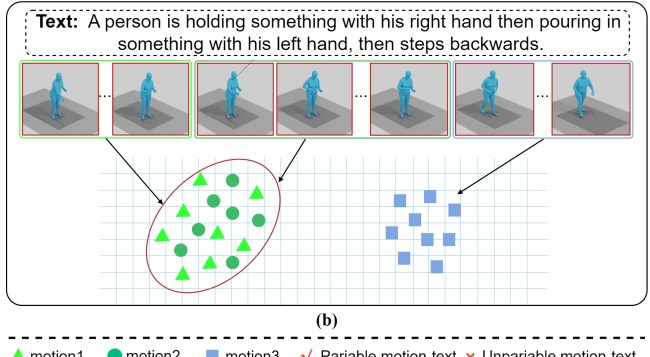

**Figure 1: Illustration of our motivations. In subfigure (a), a example retrieval results is showcased, where motion–text pairs with a coarse-grained text description that matches multiple motion sequences. In subfigure (b), motion1 and motion2 in motion sequence occur subtly and locally. These subtle motions cluster closely in the semantic space by employing simple regression methods, contributing to semantic vagueness and posing a significant challenge in achieving precise alignment.**

## 1 INTRODUCTION

Text-to-Motion Retrieval (TMR) is a significant cross-modal retrieval task, aiming at retrieving semantically similar motion sequences by the given natural language description. Recently, there has been a notable increase in research and exploration within the field of TMR. Mathis *et al.* [28] were the first to introduce the task of text to 3D human motion retrieval, establishing various evaluation benchmarks. They proposed combining motion synthesis with text-to-motion retrieval (TMR) to supervise the shared semantic space between textual descriptions and motion sequences. Nicola *et al.* [25] developed a two-stream pipeline framework for text-to-motion retrieval. Their framework incorporates a motion transformer that captures intricate motion information from skeleton data, representing motion sequences as embeddings. Additionally, they employ a pre-trained CLIP [31] text encoder to encode text as sentence embeddings. Yan *et al.* [39] introduce a method by

constructing a concise yet effective model, and design a droptriple loss to reduce semantic discrepancies in triplet training. However, these methods may have the following problems that may lead to sub-optimal matching:

• **Limitation of coarse-text description** In Fig. 1a, illustrates a retrieval results using a coarse textual description. Specifically, the Motion1 is the correct retrieval result with a similarity of 0.87, while the Motion2 and other motions (Motionk) not semantic similarity to given text, also have high similarities. These motions fail to accurately retrieval the intricate motion "thrust a javelin", simply considering it as a jump movement. Since these coarse textual descriptions may correspond to various motion sequences, training with coarse-grained texts may leads to alignment difficulties, thus affecting the accuracy of retrieval results.

• **Coordination of multiple-body parts within a motion:** In TMR, an motion is typically performed by the simultaneous movement of multiple relevant body parts. For instance, a motion sequence "A person jogs and throws an object." includes the sub-motion "throws", which may involve the movement of feet, arms, and multiple body parts working together to complete this motion. Previous methods [25, 28, 39], focusing solely on the temporal dependencies of motion sequences, falling short in capturing the essence of intricate motions. This limitation poses a challenge in obtaining a comprehensive understanding of these intricate motions.

• **Semantic vagueness between subtle motions:** In the TMR, there exists semantic vagueness between subtle motions, making it challenging to capture the semantic variations in continuous motions. As shown in Fig. 1b, the motions1 and motion2 occur successively, involving only the variation on the hands without any changes in other body parts. By aggregating these subtle motions to the semantic spaces by previous methods, we observe that these subtle motions cluster closely. Existing work [25, 28, 39] have achieved promising retrieval performance by leveraging the global embeddings, overlooking the significance of effectively capturing discriminative embeddings for a precise matching.

To address the above problems, we propose a novel solution named modal-enhanced semantic modeling method (MESM), which contain three components: the prompt-enhanced textual module (PTM), the skeleton-enhanced motion module (SMM), and the text-driven semantics interaction module (TSIM). Specifically, given a text query, we propose a prompt-enhanced textual module that utilizes a large language model to generate fine-grained text descriptions. The descriptions maintain strict chronological order while accurately specifying movements of related body parts, which comprehensively captures the fine-grained textual semantics for precise matching. Moreover, considering the human body as a graph structure, we implemented a skeleton-enhanced motion module to effectively enhance the model's capability to represent intricate motions. In this module, we employ a multi-layer graph conventional network to meticulously model the intricate spatial dependencies among relevant body parts within a motion. Finally, to effectively solve semantic vagueness between subtle motions, we introduce a text-driven semantics interaction module to obtain discriminative motion embeddings to achieve precise matching between subtle motions and text. The TSIM assigns motion features into a set of

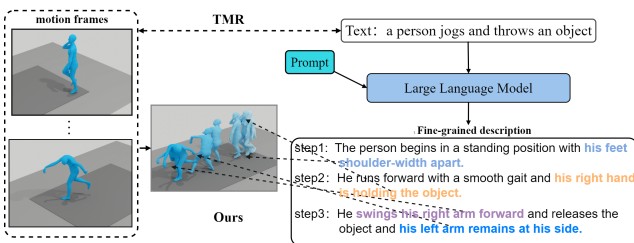

**Figure 2: In our prompt-enhanced textual module, the large language model generates a detailed text description step by step for the motion sequence. The TMR aligns the coarse-grained text and the motion sequence globally. However, we bulid a precise alignment between fine-grained descriptions and movement of relevant body part.**

local aggregated descriptors, then employ a cross-attention mechanisms to aggregate discriminative motion embeddings under the guidance of textual description, enabling precise semantic alignment between subtle motions and corresponding texts. The main contributions of this work can be summarized as follows:

- We present a novel modal-enhanced semantic modeling method to achieve fine-grained alignment by modeling the enhanced semantics information from motion and text level.
- We propose a prompt-enhanced textual module to obtain fine-grained descriptions by utilizing a large language model, which comprehensively captures the textual semantics for a precise matching.
- We design a skeleton-enhanced motion module to capture spatial dependencies among relevant body parts within a motion, which effectively enhances the model's capability to model intricate motion sequences.
- We introduce a text-driven semantics interaction module to aggregate discriminative motion embeddings conditioned by textual descriptions, which enables a precise alignment between subtle motions with corresponding texts.

## 2 RELATED WORK

### 2.1 Text-motion Retrieval

Recent advancements in research on text to 3D human motion retrieval [25, 28, 39] (TMR) have been increasingly emphasized. Compared with image retrieval [4, 30, 42], TMR is a challenging task that aims to construct a common space between text descriptions and 3D human motions by aligning the sentences and motion sequences. Mathis *et al*. [28] firstly define text to 3D human motion retrieval as an independent task. They present a method to jointly train text-to-motion retrieval and text-to-motion synthesis,by employing the contrastive learning with a special attention to the definition of negatives samples. Nicola *et al*. [25] propose a transformer-based motion encoder capture spatio-temporal informations in skeleton data and a CLIP-based [31] text encoder to obtain text embeddings. Yan *et al*. [39] propose a method by constructing a concise yet effective model, and introduce droptriple loss to minimize semantic discrepancies during triplet training. However, existing methods focus on aligning the global representations , which is inadequate to achieve precise semantics matching between intricate motion sequence and their corresponding text.

## 2.2 Prompt Learning

Prompt learning is initially introduced in the field of natural language processing (NLP) [9, 16, 19, 21, 26, 34]. Using prompt learning approaches, pretrained language models can be adapted to specific few-shot tasks by incorporating extra hand-crafted or dynamically learnable prompt tokens. Kalakonda *et al.* [13] utilize GPT-3 [3] to enhance coarse-grained descriptions. However, they apply simplistic zero-shot prompts that generate overly detailed, fine-grained texts filled with excessive, non-essential information. Athanasiou *et al.* [2] employ GPT-3 [3] to extract information about the body parts involved in various motion descriptions and then integrate these details into the corresponding motion sequences. However, these two works focus solely on single motion description and do not fully explore the extent to which large language models can comprehend the detailed of time, space, and human bodies from a coarse-grained motion description.

## 2.3 Cross-Modal Semantic Alignment

Cross-modal Semantic alignment [5, 6, 18, 23, 36–38, 40, 41, 44, 45] is a significant task in multi-modality scenarios, which aims to bring together instances from different modalities that share similar semantics. Wang *et al.* [37] present a effctive approach for the local alignment of textual and video inputs, inspired by the success of NetVLAD encoding [1], Zhu *et al.* [45] introduce the FIMA framework, which combines dense contrastive learning, foreground sampling, and a motion decoder to address weak alignment between modalities at the pixel and frame levels. Dong *et al.* [6] introduce a region-to-patch framework that contains a coarse-to-fine encoding branch to extract different granularities for a comprehensive cross-modal alignment. Wang *et al.* [38] present a hierarchical alignment framework that addresses cross-modal correspondence at various levels of granularity, achieving unified multi-grained alignment. Despite significant advancements, existing methods are typically designed for the image or video. Due to the semantic vagueness among subtle actions and complexity of the motions, it may be difficult to directly incorporate it into 3D motion scenarios.

## 3 METHODOLOGY

In this section, we provide a comprehensive overview of each component of our proposed MESM, depicted in Fig. 3. We first introduce the overall network architecture. Then, we elaborate on our proposed Prompt-Enhanced Textual Module (PTM), Skeleton-Enhanced Motion Module (SMM) and Text-Driven Semantics Interaction Module (TSIM). Finally, we detail the similarity calculation and objective function for text to 3D motion retrieval.

## 3.1 Model Overview

Unlike the previous text-motion retrieval methods [25, 28], which solely aligning the global embeddings, we introduce a novel framework capturing the enhanced fine-grained semantics to achieve a precise fine-granied alignment. Specifically, our modal consists of there modules. (1) The PTM utilizes a large language model provided appropriate prompts to generate detailed descriptions that maintain strict chronological order while specifying movements of related body parts with appropriate granularity. (2) The SMM

utilizes a graph convolutional network to learn the spatial relationships between skeleton joints as complement to piror motion fetures [10] and then employs a transformer encoder to capture the temporal dependencies within motion sequence . (3) The TSIM adopts multi-head attention to encode discriminative features conditioned by language description for precise alignment. Next, we will describe the above modules in details.

## 3.2 Prompt-Enhanced Textual Module

In real scenarios, there exists numerous coarse textual descriptions, such as "A man squats," making it difficult to effectively retrieval motions using these text description. Additionally, motion sequences consists of specific motions at relevant body parts, posing challenges in precisely aligning these intricate motion sequences with coarse-grained text. To address this issue, we utilizes a large language model, provided appropriate prompts, to generate detailed descriptions that maintain a strict chronological order while specifying movements of related body parts.

**Prompt-guided Text Learning.** We expect GPT-4.0 to properly expand text descriptions from coarse-grained to fine-grained ones, specifying movements of relevant body parts. To be more specific, an optimal fine-grained description should be in time order and specify spatial changes of relevant body parts, omit unnecessary details. After multiple empirical trials, we determine the following *Prompt* : "Provide a fine-grained textual description based on our provided coarse-textual decription, the new description should be in chronological order, step by step, and specify the movement of relevant body parts.**[example1],[example2]**". The example is a coarse textual description and its fine-grained version. By feeding prompt into LLM, we can obtain fine-grained step description $STEP_k$ corresponding to the original coarse-grained ones, which describe the movement of relevant body parts step by step. As illustrated in Fig. 2, the coarse-text is expanded into a fine-grained description by the large langnage model with our designed prompt, that precisely aligns specific body parts with description, thus ensuring a precise matching between the text and the motion sequence.

**Step-aware Self-Attention.** Initially, we utilize pre-trained Distill-BERT to extract step features for each motion step in fine-grained description. Subsequently, we aggregate the tokens of each step using an average pooling layer to generate step embeddings.

$$\mathbf{step}_{[k]} = \text{AvgPools} \left( \text{DistillBERT} \left( STEP_k \right) \right) \quad (1)$$

where $\mathbf{step}_{[k]} \epsilon \mathbb{R}^{d_t^s}$, $d_t^s$ is the dimension of step embeddings. After obtaining the step embeddings, our step-aware self-attention mechanism incorporates hard positional embeddings (**pos**) through sine and cosine functions, and feeds these position-encoded step embeddings into a multi-head self-attention. we obtain the fine-grained textual features $\mathbf{t}_i$ by

$$\mathbf{t}_s = \text{MH-Attn} \left( \mathbf{step} + \mathbf{pos} \right) \quad (2)$$

$$\mathbf{t}_i = Norm \left( FFN \left( \mathbf{t}_s \right) + \mathbf{t}_s \right) \quad (3)$$

where $FFN(\cdot)$ is a feed-forward network that consists of two linear projection layers with ReLU activations. This step-aware self-attention improves the capture of temporal relationships among steps and enhances the quality of fine-grained features through interactions among step embeddings.

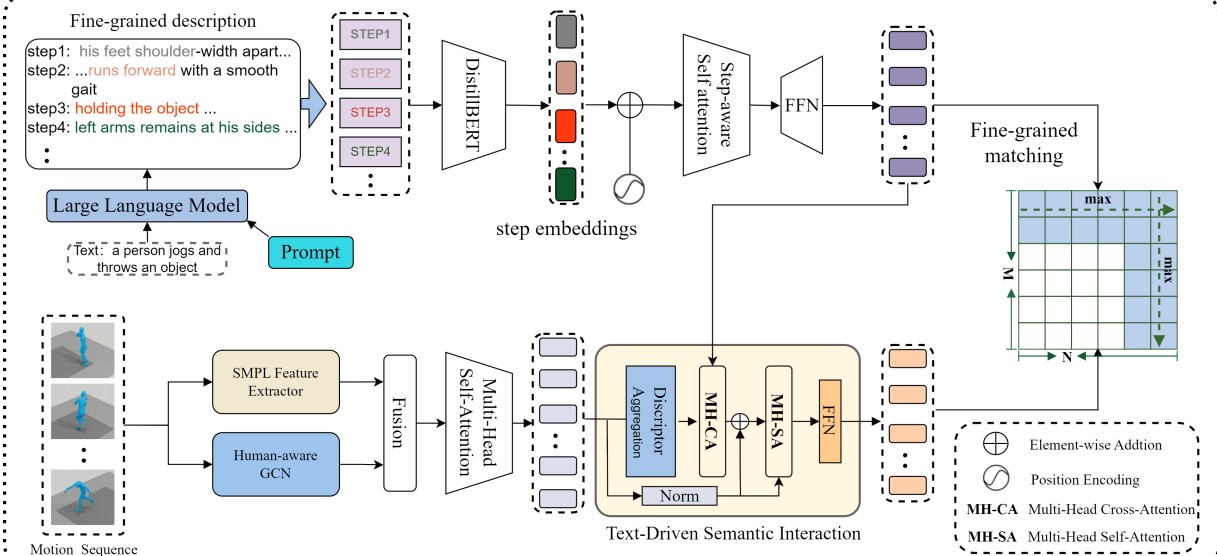

**Figure 3: The framework of our proposed MESM for text to motion retrieval. MESM consists of several key components. Firstly, we employ a large language model to expand the coarse-textual description to fine-grained ones, specifing the movement of relevant body part. And in skeleton-enhanced motion module, a human-aware GCN network is designed to enhance motion embeddings by capturing spatial dependencies among relevant body parts. Then, we build a text-driven semantics interaction module to learn discriminate motion embedding guided by text description for a precise matching. Lastly, we employ a bidirectional token-wise similarity calculation between each motion embedding and all text embeddings.**

### 3.3 Skeleton-Enhanced Motion Module.

In TMR, most motions are completed through the coordinated movement of various relevant body parts. To effectively capture the spatial relationship between these specify body parts, we introduce the Skeleton-Enhanced Motion Module(SMM) to utilizes a multi-layer Graph Convolutional Network for spatial modeling among body parts, by capturing intricate dependencies between body joints. We first employ the model from previous work [10] as the feature extractor. This feature extractor is utilized to extract frame features $\mathbf{m}_t = \left[\mathbf{m}_t^1, \mathbf{m}_t^2, \dots, \mathbf{m}_t^L\right] \in \mathbb{R}^{L \times d_m^r}$. However, due to the coordination of multiple body parts involved in intricate motions, the motion sequences extracted by using the SMPL parameter model may potentially lose some spatial information about the relationships between relevant body parts.

**Human-Aware GCN Network.** Considering the human body can be interpreted as a nature graph structure. Inspired by previous work action-recognition [14, 22, 32, 33] and pose-estimation [8, 35, 43], we can represent the human body's joints and bones as a graph $G = \{N; E\}$. Nodes are represented by $J = \{j_1, j_2, ..., j_n\}$, with $n$ indicating the number of joints. The adjacency matrix $A = (a_{ik}) \in \mathbb{R}^{J \times J}$ denotes the correlation between joint $j_i$ and $j_k$, where $a_{ik}$ is defined as:

$$a_{ik} = \begin{cases} 1, & \text{if } j_i \text{ and } j_k \text{ are connected.} \\ 0, & \text{others} \end{cases} \quad (4)$$

Specially, we employ a multi-layer Graph Convolutional Network inside each frame to extract spatial feature embedding and compute

correlations between each pair of joints in single frame independently. Hence,the output of l-layer GCN is computed by :

$$H^{(l)} = \sigma\left(\tilde{D}^{-\frac{1}{2}} \tilde{A} \tilde{D}^{-\frac{1}{2}} H^{(l-1)} W^{(l)} + b^{(l)}\right). \quad (5)$$

Here, $\tilde{A}$ is the adjacency matrix of the undirected graph G with added self-connections. $\tilde{D} = \sum_j \tilde{A}_{ij}$ and $W^{(l)}$ is a trainable weight matrix. $\sigma(\cdot)$ is an activation function. $H^{(l)} \in \mathbb{R}^{J \times D}$ is the matrix of activations in the $l^{th}$ layer. $b^{(l)}$ is a bias term. After obtaining inter-joint dependencies through our multi-layer Graph Convolutional network, we obtain motion representation $\mathbf{m}_s \in \mathbb{R}^{F \times d_m^p}$ by a flatten operation.

Lastly, based on the features extracted by feature extractor and the spatial motion features $\mathbf{m}_s$, we employ a transformer encoder to obtain the final motion representation:

$$\mathbf{m}_l = TransEncoder(MLP(\mathbf{m}_t + FC(\mathbf{m}_s))) \quad (6)$$

Here, $FC(\cdot)$ is a linear layer, mapping $\mathbf{m}_s$ from dimension $d_m^p$ to $d_m^s$. A transformer encoder applied to the sum of $\mathbf{m}_t$ and $\mathbf{m}_s$ enhances the motion embeddings, providing more comprehensive semantic for intricate motion.

### 3.4 Text-Driven Semantics Interaction Module

In real-world scenarios, most motion variations occur locally and subtly. There exists semantic vagueness between these local and subtle motions, making it challenging to achieve precise alignment between these motions and text. To effectively capture the discriminative motion embeddings, we introduce a Text-Driven

Semantics Interaction Module (TSIM) to aggregate motion representations under the guidance of textual embeddings, enabling a precise semantics alignment.

**Motion Descriptor Aggregation.** We utilize a motion-wise descriptors aggregation method to capture local motion features. We first build a descriptors set with $k$ motion-wise cluster descriptors $\{p_1, \ldots, p_K\} \in \mathbb{R}^{d_p}$. Then, the motion frame features $\mathbf{m}_l$ could be assigned to the motion-wise descriptors , the $j$-th motion-wise descriptors is obtained as followed:

$$m_p^j = \left\| \sum_{i=1}^{B} \frac{\exp\left(m_l^j p_j^\top + b_j\right)}{\sum_{k=1}^{K} \exp\left(m_l^j p_k^\top + b_k\right)} \left(m_l^j - p'_k\right) \right\|_2 \quad (7)$$

where $p'_k$ has same size with $p_k$ as a trainable weight, $b_j$ is a learnable bias, and $\|\cdot\|_2$ indicates the L2 normalization. The motion information within the motion sequence is described using the motion-wise descriptors. After assigning each frame feature to the descriptors, we could obtain the descriptors-based motion local features $\mathbf{m}_p = \left\{\mathbf{m}_p^1, \mathbf{m}_p^2, \ldots, \mathbf{m}_p^k\right\}$, where $\mathbf{m}_p \in \mathbb{R}^{k \times d_c}$.

**Text-Guided Cross-Modal Interaction.** As shown in Fig. 3, to enable the interaction between two modalities, we first apply a cross-modal attention to obtain the linguistic representations $\mathbf{t}_p$ corresponding to the motion local representations $\mathbf{m}_p$.

$$\text{Cross-Attn}\,(Q_m, K_t, V_t) = \text{softmax}\left(\frac{Q_m K_t^T}{\sqrt{d_c}}\right) V_t \quad (8)$$

$$\mathbf{t}_p = Norm\,(\text{Cross-Attn}\,(Q_m, K_t, V_t)\,W_O) \quad (9)$$

where $Q_m, K_t, V_t$ represent the query, key, and value obtained by projecting the motion and textual features. Norm denotes L2 normalization. The parameter matrix $W_O$ is a projection weight matrix.

Based on $t_p$, we employ another multi-head attention module to perform language-conditioned self-attention for the discriminative motion features. Specifically, we take the sum of $t_p$ and $m_p$ as the query and key inputs for the multi-head attention module, facilitating the computation of feature correlations in both the motion and linguistic representations. The attention weight value from position i to position j is formulated as follows:

$$\begin{cases} Q' = W_Q^\top (\mathbf{m}_p + \mathbf{t}_p) \\ K' = W_K^\top (\mathbf{m}_p + \mathbf{t}_p) \\ \text{attn}_{i,j} = \text{Softmax}\left(\frac{Q'(i)^\top (K'(j) + W_K^\top R(i-j))}{\sqrt{d_k}}\right) \end{cases} \quad (10)$$

where $W_Q$ and $W_K$ are the linear projection weights for the query and key, and $R(\cdot)$ represents the positional encodings of relative positions. This text-guided cross-modal interaction enables the model to capture discriminative features for subtle motions based on the given textual descriptions. The multi-head attention outputs final motion representations as $\mathbf{m}_i = \left\{\mathbf{m}_i^1, \mathbf{m}_i^2, \ldots, \mathbf{m}_i^L\right\} \in \mathbb{R}^{L \times d_c}$, $L$ is the number of final motion embeddings.

## 3.5 Similarity Calculation and Loss Function

**Similarity Calculation.** The similarity score $S(m_i, t_j)$ measures the semantic similarity between the text $t_j$ and motion sequence $m_i$.

It is computed as the mean of the maximum similarities between each motion embedding and all text embeddings in bi-directions.

$$S\,(m_i, t_i) = \frac{1}{2}\left(\sum_{n=1}^{N} \max_{m=1}^{M} \langle \mathbf{m}_i^n, \mathbf{t}_i^m \rangle + \sum_{m=1}^{M} \max_{n=1}^{N} \langle \mathbf{m}_i^n, \mathbf{t}_i^m \rangle\right), \quad (11)$$

where $M, N$ denote visual embedding and text embedding number in the $i$ th sample pair.

**Loss Function.** Many studies [7, 17, 20] have proved that not only do positive samples learn joint semantics from visual-textual alignment, but a sufficient number of negative samples better fill up the semantic inequality. In our study, we adopt contrastive learning along with a hinge-based triplet loss [12], deliberately establishing distinct threshold based on relative similarity instead of absolute similarity. This formulation is expressed as follows:

$$\mathcal{L}_{\text{neg}} = \frac{1}{N} \sum_{(m,t) \in B} [S(m, \hat{t}) - S(m, t) + w]_+ \quad (12)$$

$$+ [S(\hat{m}, t) - S(m, t) + w]_+, \quad (13)$$

Here, $[\cdot]_+ = \max(x, 0)$. $S(m, t)$ represents the similarity between positive samples and $w$ is a boundary factor. $B$ is the mini-batch. $S(m, \hat{t})$ represents the similarity between $m$ and the hardest textual negative sample corresponding to $m$ (same as $S(\hat{m}, t)$ ).

Additionally, we employ a logistic loss function to strengthen the correlations among positives samples, which is denoted as follows:

$$\mathcal{L}_{\text{pos}} = \frac{1}{N} \sum_{(m,t) \in N} \log\left[1 + e^{-\tau S(m,t)}\right] \quad (14)$$

Finally, our total loss function is designed with the linear combination of $\mathcal{L}_{\text{pos}}$ and $\mathcal{L}_{\text{neg}}$ :

$$\mathcal{L} = \mathcal{L}_{\text{pos}} + \alpha \cdot \mathcal{L}_{\text{neg}} \quad (15)$$

## 4 EXPERIMENTS

### 4.1 Datasets

To validate the proposed methods, we employ two recently widely used 3D human motion datasets, HumanML3D [10] and KIT Motion Language [29] .

**HumanML3D** [10]is currently the largest 3D human motion dataset accompanied by textual descriptions. The motion sequences are sourced from two well-established and widely-used motion-capture datasets: AMASS [24] and HumanAct12 [11]. Following the setup in the benchmark [28], the dataset is divided into training, validation, and test sets, containing 23,384, 1,460, and 4,380 motion sequences, respectively. Each motion sequence is paired with around three text descriptions of varying lengths.

**KIT Motion-Language** [29] contains 3,911 recordings of full-body motion paired with 6,278 text descriptions. Each motion sequence is described in one to four texts, with an average description length of approximately 8 words. Consistent with the benchmark (TMR) setup, we utilize 4,888, 300, and 800 motion sequences for the training, validation, and test sets, respectively.

### 4.2 Implementation Details

All experiments are conducted on a workstation with NVIDIA A100 GPUs , by using the PyTorch-1.10 library. We set the mini-batch size as 64 and employ the Adam optimizer [15] to optimize our model

**Table 1: Performance comparison with the state-of-the-art methods on HumanML3D [10]. The best results are shown in bold.**

| Settings | Methods | Text-to-Motion | | | | | | Motion-to-Text | | | | | | Rsum ↑ |
|---|---|---|---|---|---|---|---|---|---|---|---|---|---|---|
| | | R@1 ↑ | R@2 ↑ | R@3 ↑ | R@5 ↑ | R@10 ↑ | MedR ↓ | R@1 ↑ | R@2 ↑ | R@3 ↑ | R@5 ↑ | R@10 ↑ | MedR ↓ | |
| (a) Normal | TSA [10] | 1.80 | 3.42 | 4.79 | 7.12 | 12.47 | 81.00 | 2.92 | 3.74 | 6.00 | 8.36 | 12.95 | 81.50 | 63.57 |
| | TEMOS [27] | 2.12 | 4.09 | 5.87 | 8.26 | 13.52 | 173.00 | 3.86 | 4.54 | 6.94 | 9.38 | 14.00 | 183.25 | 72.58 |
| | DTL [39] | 2.30 | 4.63 | 6.66 | 10.06 | 16.40 | 76.00 | 2.64 | 4.91 | 7.34 | 10.95 | 17.21 | 76.00 | 83.10 |
| | MoT [25] | 2.61 | 4.72 | 6.90 | 10.66 | 17.79 | 60.00 | 4.03 | 5.07 | 7.43 | 11.23 | 17.68 | 64.25 | 88.12 |
| | TMR [28] | 5.68 | 10.59 | 14.04 | 20.34 | 30.94 | 28.00 | 9.95 | 12.44 | 17.95 | 23.56 | 32.69 | 28.50 | 178.18 |
| | MESM (Ours) | **7.16** | **12.52** | **16.70** | **24.22** | **35.38** | **23.00** | **11.19** | **13.81** | **19.59** | **25.96** | **35.93** | **23.25** | **202.46** |
| (b) Threshold | TSA [10] | 5.30 | 7.83 | 10.75 | 14.59 | 22.51 | 54.00 | 4.95 | 5.68 | 8.93 | 11.64 | 16.94 | 69.50 | 109.12 |
| | TEMOS [27] | 5.21 | 8.22 | 11.14 | 15.09 | 22.12 | 79.00 | 5.48 | 6.19 | 9.00 | 12.01 | 17.10 | 129.00 | 111.56 |
| | DTL [39] | 5.32 | 8.11 | 12.27 | 17.63 | 26.85 | 41.00 | 6.01 | 7.23 | 10.55 | 14.08 | 20.98 | 71.50 | 129.03 |
| | MoT [25] | 5.11 | 8.65 | 12.45 | 18.07 | 28.51 | 35.00 | 6.57 | 8.01 | 11.47 | 15.62 | 22.70 | 51.25 | 137.16 |
| | TMR [28] | 11.60 | 15.39 | 20.50 | 27.72 | 38.52 | 19.00 | 13.20 | 15.73 | 22.03 | 27.65 | 37.63 | 21.50 | 229.97 |
| | MESM (Ours) | **14.28** | **18.86** | **24.48** | **30.84** | **42.97** | **15.00** | **14.17** | **16.99** | **23.91** | **30.22** | **40.17** | **17.50** | **256.89** |

**Table 2: Performance comparison with the state-of-the-art methods on KIT-ML [29].The best results are shown in bold.**

| Settings | Methods | Text-to-Motion | | | | | | Motion-to-Text | | | | | | Rsum ↑ |
|---|---|---|---|---|---|---|---|---|---|---|---|---|---|---|
| | | R@1 ↑ | R@2 ↑ | R@3 ↑ | R@5 ↑ | R@10 ↑ | MedR ↓ | R@1 ↑ | R@2 ↑ | R@3 ↑ | R@5 ↑ | R@10 ↑ | MedR ↓ | |
| (a) Normal | TSA [10] | 3.37 | 6.99 | 10.84 | 16.87 | 27.71 | 28.00 | 4.94 | 6.51 | 10.72 | 16.14 | 25.30 | 28.50 | 129.39 |
| | TEMOS [27] | 7.11 | 13.25 | 17.59 | 24.10 | 35.66 | 24.00 | 11.69 | 15.30 | 20.12 | 26.63 | 36.39 | 26.50 | 207.84 |
| | DTL [39] | 6.77 | 13.28 | 16.67 | 23.18 | 37.24 | 18.00 | 9.11 | 14.32 | 20.31 | 25.26 | 38.02 | 18.00 | 204.16 |
| | MoT [25] | 6.23 | 11.07 | 16.54 | 23.92 | 37.15 | 20.00 | 10.56 | 13.49 | 20.61 | 27.61 | 38.04 | 19.50 | 205.22 |
| | TMR [28] | 7.23 | 13.98 | 20.36 | 28.31 | 40.12 | 17.00 | 11.20 | 13.86 | 20.12 | 28.07 | 38.55 | 18.00 | 221.80 |
| | MESM (Ours) | **9.29** | **17.05** | **22.31** | **29.13** | **41.02** | **16.00** | **12.75** | **16.41** | **24.17** | **32.59** | **42.88** | **15.50** | **247.60** |
| (b) Threshold | TSA [10] | 13.25 | 22.65 | 29.76 | 39.04 | 49.52 | 11.00 | 10.48 | 13.98 | 20.48 | 27.95 | 38.55 | 17.25 | 265.66 |
| | TEMOS [27] | 18.55 | 24.34 | 30.84 | 42.29 | 56.39 | 7.00 | 17.71 | 22.41 | 28.80 | 35.42 | 47.11 | 13.25 | 323.86 |
| | DTL [39] | 21.72 | 31.25 | 38.12 | 46.19 | 61.85 | 7.00 | 18.17 | 22.15 | 30.56 | 40.78 | 50.12 | 10.00 | 360.91 |
| | MoT [25] | 20.87 | 30.92 | 38.17 | 47.58 | 60.05 | 6.00 | 18.45 | 23.41 | 31.17 | 41.22 | 50.89 | 10.25 | 362.73 |
| | TMR [28] | 24.58 | 30.24 | 41.93 | 50.48 | 60.36 | 5.00 | 19.64 | 23.73 | 32.53 | 41.20 | 53.01 | 9.50 | 377.70 |
| | MESM (Ours) | **26.38** | **32.96** | **42.87** | **51.64** | **61.36** | **3.00** | **22.01** | **25.57** | **32.26** | **42.49** | **55.73** | **7.25** | **393.27** |

. To prevent overfitting, we utilize a learning rate of 0.0002 and initiate decay of 15% every 10 epochs after epoch 30. For the both HumanML3D and KIT-ML datasets, we set the maximum motion sequence length and training epochs to 200 and 300, respectively. In addtion, the hyper-parameters $w$ in Eq. 12 is set to 0.2, $\tau$ in Eq. 14 is 10, and the trade-off weight $\alpha$ in the overall loss function is set to 1.0. The representation dimension $d_c$ of common space is set to 256. To ensure a consistent comparison with the baseline, we follow the experimental settings presented in previous work [28], where a text is randomly selected as the matching text for training, and the first text in the test set is used to report the evaluation performance under both the "Normal" and "Threshold" setups. Under the "Normal" setup, all test sets were evaluated without modification. Under the "Threshold" setup, we searched all test sets, considering a motion correct if its text label closely matched the query text above a threshold (set to 0.95, following prior work [28]). We adopt the common metrics to report retrieval performance, including Recall at K (R@K), Median Rank (MedR), and Rsum, following the benchmark [28]. The best evaluation results are highlighted in "**bold**".

## 4.3 Performance Comparison

In this section, we compare our proposed method with the other state-of-the-arts methods on various text-to-motion retrieval, including TEMOS [27], TSA [10], TMR [28], DTL [39] and MoT [25]. Considering that MoT [25] and DTL [39] differ from our experimental settings, we re-implement them on our settings and report the results based on official codes. Tab. 1 and Tab. 2 provide a comparative overview of our model against state-of-the-art methods on the HumanML3D and KIT datasets, respectively. We observe that our modal achieves better performance than existing methods on most of the metrics on both text-to-motion and motion-to-text retrieval settings. Specifically, under "Normal" setups, it outperforms the current state-of-the-art TMR [28], achieving relative improvement of 1.48% and 2.06% on the R@1 metric for text-to-motion retrieval on both HumanML3D and KIT datasets . Furthermore, the overall retrieval quality, as indicated by the Rsum metric, experiences a significant boost (+24.28%, +25.80%) across both datasets.

Under the "Threshold" setups, our proposed MESM continues to outperform all baselines, particularly the TMR [28], which incorporates specific designs for negative filtering. The effectiveness of our

methods demonstrates that enhanced visual and textual features provide richer semantic information for the retrieval process.

## 4.4 Ablation Study

**The effectiveness of each component.** In Tab. 3, we conduct a thorough ablation study to verify the effectiveness of proposed components in our methods. In the first group, only the single component is used for enhancing the retrieval process. We observe that 1) the first row of the first group shows our baseline that performs achieves R@1 of 5.48 slightly below TMR [28]. 2) These three components show a better retrieval performance than baseline. It indicates that PTM and SMM can enhance our obtained features to achieve fine-grained alignment between text and motion sequences. 3) Only TSIM used in our methods achieves the best performance in comparison with other components. It proves that TSIM can learn discriminative motion embeddings for precise text-motion alignment to improve retrieval performance. For the second group of ablation studies, we pair different components to investigate their interrelated effects. We observe that 1) compared to using a single component , combining components leads to a notable improvement in retrieval performance. It indicates that the proposed PTM, SMM and TSIM can provide varying fine-grained semantics and are complementary to each other. 2) The most significant improvement is achieved in the integration of PTM and SMM . It may be attributed that the SMM and PTM could provide more richer semantics to enhance representations. 3) The integration of PTM, SMM, and TSIM results in a further boost to retrieval performance and achieves the state-of-the-art. This reveals that the proposed MESM framework captures the comprehensive semantics to achieve precise retrieval performance.

**Generalization Analysis of PTM and SMM.** In this subsection, we aim to verify the generalization of our proposed prompt-enhanced textual module (PTM) and skeleton-enhanced motion module (SMM). We plug the proposed PTM and SMM into the different backbones TEMOS, MoT, and TMR (denoted as "+ PSM*") on the HumanML3D dataset. The improvement performance by our PSM* are marked in green. The results in shown in Tab. 4. Specifically, R@1 of text-motion matching on MoT [25] obtains a significant boost of 3.26% . It may be attributed to the backbone of MoT [25] only focusing on the coarse-grained matching while our PSM* enhances the semantic information both on text and motion. However, the improvement in results for TMR is limited, with R@1 increasing by only 1.13%. This may be attributed to the complex distribution alignment and motion synthesis within TMR, both of which could offer supplementary information for auxiliary alignment between text and motion. All baselines show a satisfying improvement by equipping with our PTM and SMM, which demonstrate the generalization of our proposed components.

**The ablation study of descriptor number.** We conduct ablation study on the number of motion descriptors as proposed in our text-driven semantics interaction module. Based on the average movement number within the sequence, we set the descriptor numbers from 3 to 6 to explore the optimal setting on HumanML3D and KIT-ML. We report the Rsum to measure the overall performance. As shown in Tab. 5, it demonstrates that a larger setup of descriptor numbers brings much retrieval performance. However, an increase beyond four descriptors tends to degrade performance.

**Table 3: The ablation studies of the proposed components in our method. We report the experimental results on the text-to-motion retrieval on both HumanML3D and KIT-ML datasets.**

| PTM | SMM | TSIM | HumanML3D | | | KIT-ML | | |
|-----|-----|------|-----------|-----------|-----------|-----------|-----------|-----------|
| | | | R@1 ↑ | R@2 ↑ | R@3 ↑ | R@1 ↑ | R@2 ↑ | R@3 ↑ |
| ✗ | ✗ | ✗ | 5.48 | 10.46 | 14.02 | 6.88 | 13.58 | 19.76 |
| ✓ | ✗ | ✗ | 6.12 | 10.97 | 14.34 | 7.87 | 14.97 | 20.76 |
| ✗ | ✓ | ✗ | 5.92 | 11.54 | 15.07 | 7.59 | 15.01 | 20.98 |
| ✗ | ✗ | ✓ | 6.09 | 11.67 | 15.24 | 8.02 | 15.65 | 21.08 |
| ✓ | ✓ | ✗ | 6.72 | 12.13 | 16.21 | 8.48 | 16.43 | 21.67 |
| ✗ | ✓ | ✓ | 6.24 | 11.38 | 15.82 | 8.19 | 15.98 | 21.52 |
| ✓ | ✗ | ✓ | 6.49 | 11.87 | 15.76 | 8.17 | 16.02 | 21.25 |
| ✓ | ✓ | ✓ | **7.16** | **12.52** | **16.70** | **9.29** | **17.05** | **22.31** |

**Table 4: Generalization study of the proposed PTM and SMM on the HumanML3D dataset. The PSM* indicates only incorporating the prompt-enhanced textual module and skeleton-enhanced motion module into the methods. The performance improvement compared with base backbones achieved by PSM* is marked in green.**

| Methods | Text-to-Motion | | |
|---------|-----------|-----------|-----------|
| | R@1 ↑ | R@2 ↑ | R@3 ↑ |
| TEMOS [27] | 2.12 | 4.09 | 5.87 |
| + PSM* | $5.32_{\uparrow 3.20}$ | $8.89_{\uparrow 4.80}$ | $13.01_{\uparrow 7.14}$ |
| MoT [25] | 2.61 | 4.72 | 6.90 |
| + PSM* | $5.87_{\uparrow 3.26}$ | $10.63_{\uparrow 5.91}$ | $15.06_{\uparrow 8.16}$ |
| TMR [28] | 5.68 | 10.59 | 14.04 |
| + PSM* | $6.81_{1.13\uparrow}$ | $12.07_{1.48\uparrow}$ | $16.31_{\uparrow 2.27}$ |

It may be attributed that excessively numerous descriptors can introduce redundant semantics and lead to dispersed semantics at the descriptors to damage the fine-grained alignment.

**The Impact of hyper parameters.** We conduct a group of experiments with different values of the boundary factor $w$ in Eq. 12 and weights $\alpha$ in total loss function. As shown in Fig. 5, we set the $w$ from 0.1 to 0.5, and $\alpha$ from 1e-2 to 1e+2. In Fig. 5a, we get the best retrieval performance when $w = 0.2$. We note that as $w$ increases to 0.5, there is a substantial decrease in retrieval performance across all metrics. This could be because that the model might overly penalize negative sample pairs that are closely similar to positive ones,if the margin value is too large. Furthermore, in Fig. 5b, the overall retrieval performance initially increased before reaching saturation at $\alpha = 1.0$ and then slightly declined. These results suggest that a larger value of $\alpha$ may lead the model to assign excessive weight to hard negative pairs, while a smaller value of $\alpha$ may underestimate the impact of hard negative pairs.

## 4.5 Visualization Results

In Fig.4, we present the visualization of the retrieval results obtained by our proposed MESM (Ours), as well as the baseline approaches TMR [28] and MoT [25] on the HumanML3D dataset. Successful retrieval results are highlighted with a green border. For the first

**Query:** The man walked up to the door and knocked on it.

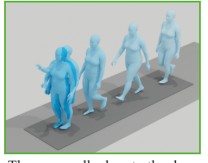 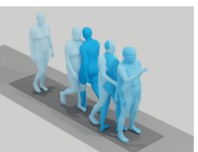 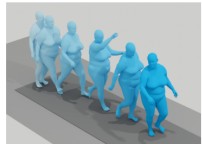 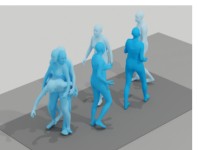 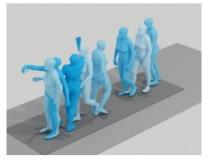 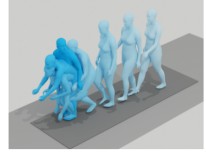

The man walked up to the door and knocked on it.

Person walks forward, raises right hand and waves, then turns and walks back.

Person walks six steps to side with right arm up

A person walk forward, picks something up, then walks back while tossing it up.

A person walks forward, then grabs something with right arm ,then walks back.

Walking forward and then bending down.

MESM(Ours)                                        TMR                                        MoT

**Query:** A person walks forward, and puts things together and begins to stir them .

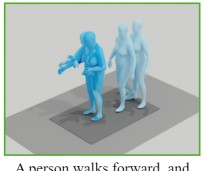 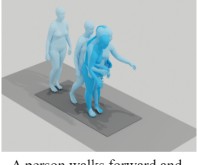 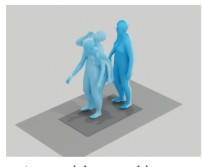 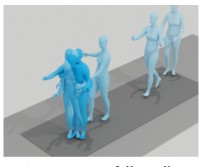 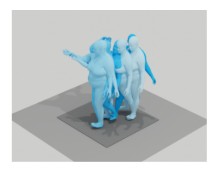 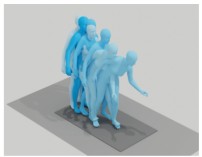

A person walks forward, and puts things together and begins to stir them .

A person walks forward and picks things up and puts them down with their hands.

A man picks something up, drinks it and then puts it back.

A person gracefully walks forward, picks up an object, and raises it towards their face.

The person walks forward, waved their hand then stepped backward.

A person walks forwards, grabs something with his left hand, and walks backwards.

MESM(Ours)                                        TMR                                        MoT

**Figure 4: Visualization of the retrieval results. We present the top-2 motion sequences retrieved by our proposed method MESM , as well as by TMR [28] and MoT [25]. The text below each motion sequence represents the ground truth. The successful retrieval results are highlighted by the green border.**

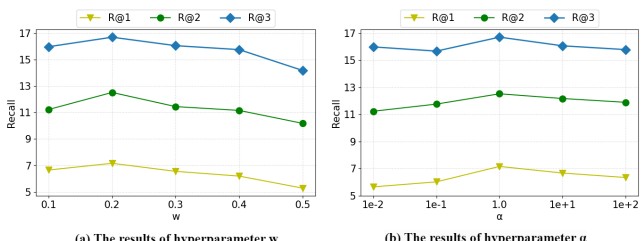

(a) The results of hyperparameter $w$                (b) The results of hyperparameter $\alpha$

**Figure 5: Hyperparameters analysis of the $w$ and $\alpha$. The results are reported on HumanML3D at the text-to-motion retrieval in subfigure (a) and subfigure (b), respectively.**

**Table 5: Parameter analysis for the number of descriptors. We report the overall performance measured by Rsum on the HumanML3D and KIT-ML, respectively.**

| $K$ | Datasets | |
|---|---|---|
| | HumanML3D | KIT-ML |
| 3 | 198.49 | 241.47 |
| 4 | **202.46** | **247.60** |
| 5 | 199.17 | 240.18 |
| 6 | 197.91 | 235.62 |

example, our proposed method successfully retrieves the motion sequence. The second retrieved motion sequence also includes the key motions mentioned in the query text, such as "walks" and "knocked". Notably, the "knocked" motion sequence contains multiple coordinated movements of relevant body parts, such as "arms" and "hand", demonstrating that our method can achieve fine-grained semantics alignment by capturing intricate details within the motion sequences. In contrast, the baselines TMR [28] and MoT [25] are able to capture simple motions like "walks". However, they struggle to effectively retrieve more complex motions like "knocked",

which require coordinated movements of multiple body parts. This limitation can be attributed to their coarse-level modeling of text and motion representations, which makes it challenging to align intricate motions with textual descriptions.

For the second example, our proposed method achieves successful retrieval of ground truth, despite the motions being subtle and indiscriminative. This demonstrates our method could effectively align these subtle motions with the text descriptions and achieves a accurate retrieval. However, both TMR and MoT fail to retrieve the ground truth motion sequences, particularly the second retrieved results. These results primarily contain minor motions that lack relevant semantics. This limitation arises from the the coarse global representation alignment that aligns the average semantics of the query with the motion sequence, which can not provide substantial meaningful semantic information.

## 5 CONCLUSION

In this paper, we propose a modal-enhanced semantic modeling method to model and align the precise semantics for text to 3D human motion retrieval. We employ prompt-enhanced textual module and skeleton-enhanced motion module to enhance the semantic information of both the text and motion modalities. In addition, we introduce a Text-Driven Semantic Interaction Module to learn more discriminative motion embeddings for precise fine-grained semantic alignment between subtle motions and the corresponding texts. In the future, we aim to design a framework that progressively models semantic information at different levels to achieve hierarchical alignment. We believe that the the findings and solutions proposed in this paper have the potential to offer valuable insights to related domains, contributing to a deeper understanding of the field.

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
