# OpenReview forum: "Modal-Enhanced Semantic Modeling for Fine-Grained 3D Human Motion Retrieval"
_acmmm.org/ACMMM/2024/Conference — MM2024 Poster_

### Official Review · Reviewer_M2KD · 2024-05-14

**Rating:** 4
**Confidence:** 3

**Summary:**

The authors proposed a framework for text to motion retrieval by designing a novel modal-enhanced semantic modeling method. The authors proposed a prompt-enhanced textual module to obtain fine-grained descriptions by utilizing a large language model. Extensive experiments and analyses on benchmark datasets demonstrate that the proposed methods achieve the state-of-the-art performance.

**Strengths:**

1. Numerous experiments have been conducted to demonstrate the efficacy of the method.
2. LLM offers a novel perspective for the text to motion retrieval.
3. The writing style and structure of the paper make it easy for readers to comprehend the methodology and significance of the study.

**Limitations:**

1. Your approach appears to be more intricate than the TMR[1].  Comparing the training and test times with other retrieval methods would be beneficial.  Providing the specific time measurements would be valuable.
2. The incorporation of LLM may entail the integration of external information, raising questions about its fairness when compared to other methods.
3. The experiment lacks part experiment compared with TMR[1] such as "dissimilar subset'' and "small batches'' in two datasets.
4. The extra visual feature extractor may raise questions about its fairness when compared to other methods.

[1]TMR: Text-to-Motion Retrieval Using Contrastive 3D Human Motion Synthesis

Further Comments

I hope the author takes the rebuttal seriously, and I will revise the review as appropriate based on the response provided in the rebuttal.

**Suitability:**

3

---

### Official Review · Reviewer_JZmZ · 2024-05-24

**Rating:** 4
**Confidence:** 3

**Summary:**

The authors propose a modal-enhanced semantic modeling method to model and align the precise semantics for text to 3D human motion retrieval. Specifically, the authors (1) employ a prompt-enhanced textual module and a skeleton-enhanced motion module to enhance the semantic information; and (2) introduce a Text-Driven Semantic Interaction Module to learn more discriminative motion embeddings.

**Strengths:**

The experiments of this paper are rich.
The motivation of this paper is clear.

**Limitations:**

(1) The overview of this paper is somewhat lacking novelty. Specifically, this paper utilizes LLM for fine-grained text expansion and designs a multimodal feature alignment module. This requires further explanation of the contributions of the authors.
(2) The caption of Figure 2 is redundant, such as "The TMR... However, we build ...". Therefore, it needs to be refined.
(3) In the abstract, the motivation of this paper needs to be refined. For example, in stating the task challenge, we can see these descriptions "However, simple global embeddings. In addition, ... ".
(4) Some paper lacks citations, such as GPT-4.

**Suitability:**

3

---

### Official Review · Reviewer_Xkrv · 2024-05-24

**Rating:** 5
**Confidence:** 3

**Summary:**

The authors introduce a fine-grained approach to the emerging text-motion retrieval task. Specifically, they employ an LLM to obtain fine-grained motion descriptions, and design an architecture able to cope with fine-grained text-motion associations. The architecture is composed of a fine-grained motion encoder, a text encoder that processes the different motion description steps extracted from the LLM, and a text-driven semantic interaction module that combines the two modalities at a fine-grained level using cross-attention. The results are reported on two well-known datasets for text-motion retrieval, and the authors perform an ablation study to validate all the proposed components.

**Strengths:**

- The paper is clearly written and its motivation is sound. Many works tackled the fine-grained matching task in the text-to-image scenario. However, the role of fine-grained matching was underexplored in this emerging text-motion retrieval task.
- The results are sound and achieve a great improvement margin over the previous coarse-grained approaches.
- The ablation studies show the effectiveness of all the proposed components.

**Limitations:**

- It is not very clear how LLM can transform a coarse-grained description into a fine-grained one without adding some expected noise, as it adds some details that may not be present or incorrect for the corresponding motion. In this sense, I was expecting some ablation experiments where the authors used the original captions and not the augmented ones. In that case, as in many other fine-grained image-text matching works, the authors could have used the word embeddings in output from DistillBERT as fine-grained text tokens (so one token for each word instead of one token for each step description).
- Concerning the related work, it is not clear why the authors included a Prompt Learning section, given that it seems they are not learning additional prompt tokens in the end-to-end training (the LLM is prompted using a fixed human-written prompt).
- The employed loss function is greatly inspired by previous fine-grained alignment methods on text and images (e.g., [1, 2]), which should be carefully cited in the Related Work section or Section 3.5.
- There are some imprecisions in the employed formalisms. For example: i) it is not clear what p’_k is in Eq. 7; in the same equation, I believe there are some incorrect indexes being used, as i is defined in the summation but never used; ii) it seems the output from the text-driven semantic interaction module should be a set of K vectors, but in line 516 instead L (the initial number of motion features before the aggregation) is reported; iii) both N and B are used to indicate the batch size (for example, Eq. 12, the summation is over B, while in Eq. 14 it is over N). This should be uniform.
- The role of Lpos is not too clear, given that the triplet loss is usually already sufficient (here called Lneg). I would have expected an ablation experiment with only Lneg.

Minor issues:
- The text is full of annoying typos that may confuse the reader. For example:
  - “to accurately retrieval” -> “to accurately retrieve”
  - “graph conventional network” -> “graph convolutional network”
  - “our modal consists” -> “our model consists”
  - “piror motion features” -> ?
- In the second paragraph of Section 4.4, the boosts are expressed in percentages. However, they seem to be absolute recall values rather than a relative increase. Please remove the “%” sign or convert them to a percentual rise/decrease.

[1] Karpathy, Andrej, and Li Fei-Fei. "Deep visual-semantic alignments for generating image descriptions." Proceedings of the IEEE conference on computer vision and pattern recognition. 2015.

[2] Lee, Kuang-Huei, et al. "Stacked cross attention for image-text matching." Proceedings of the European conference on computer vision (ECCV). 2018.

**Suitability:**

3

---

### Official Review · Reviewer_5Tsk · 2024-05-26

**Rating:** 4
**Confidence:** 3

**Summary:**

This paper proposes a modal-enhanced semantic modeling method to model (MESM) and align the precise semantics for text to 3D human motion retrieval. The proposed MESM contains three components: the prompt-enhanced textual module (PTM), the skeleton-enhanced motion module (SMM), and the text-driven semantics interaction module (TSIM), enabling precise semantic alignment between subtle motions and corresponding texts.

**Strengths:**

1. This paper proposes a prompt-enhanced textual module that utilizes a large language model (GPT-4) to generate fine-grained text descriptions, which maintain strict chronological order while accurately specifying movements of related body parts.
2. This paper introduces a text-driven semantics interaction module to aggregate discriminative motion embeddings conditioned by textual descriptions, which enables a precise alignment between subtle motions with corresponding texts.

**Limitations:**

1. How can the model eliminate situations where the generated descriptions do not match the actual 3D actions, as the descriptions generated by LLM may not always fully match the images.
2. The description of "Motion Descriptor Aggregation" in the paper is insufficient, which raises doubts about the necessity of this module.
3. For the proposed $L_{pos}$, ablation experiments are required to verify its effectiveness.
4. Some expressions in the article are redundant and need to be streamlined.

**Suitability:**

3

---

### Meta-Review · Area_Chair_j6LM · 2024-07-02

**Recommendation:** Accept (Poster)
**Confidence:** 4

**Metareview:**

There is a consensus to accept this paper.
The authors provided relevant and satisfactory responses to the reviewers' comments in their rebuttal.
The reviewer appreciated the precise alignment between subtle motions with corresponding texts, clarity, performance improvement margin, ablation studies, and richness of the experiments.